# Diagnostic Ability of Endoscopic Ultrasound-Guided Tissue Acquisition Using 19-Gauge Fine-Needle Biopsy Needle for Abdominal Lesions

**DOI:** 10.3390/diagnostics13030450

**Published:** 2023-01-26

**Authors:** Kotaro Takeshita, Susumu Hijioka, Yoshikuni Nagashio, Yuta Maruki, Yuki Kawasaki, Kosuke Maehara, Yumi Murashima, Mao Okada, Go Ikeda, Natsumi Yamada, Tetsuro Takasaki, Daiki Agarie, Hidenobu Hara, Yuya Hagiwara, Kohei Okamoto, Daiki Yamashige, Akihiro Ohba, Shunsuke Kondo, Chigusa Morizane, Hideki Ueno, Yutaka Saito, Yuichiro Ohe, Takuji Okusaka

**Affiliations:** 1Department of Hepatobiliary and Pancreatic Oncology, National Cancer Center Hospital, Tokyo 104-0045, Japan; 2Cancer Medicine, Jikei University Graduate School of Medicine, Tokyo 105-0003, Japan; 3Endoscopy Division, National Cancer Center Hospital, Tokyo 104-0045, Japan; 4Department of Thoracic Oncology, National Cancer Center Hospital, Tokyo 104-0045, Japan

**Keywords:** endoscopic ultrasound-guided tissue acquisition, 19-gauge needle, diagnostic ability, fine-needle biopsy, Franseen needle, pancreatic cancer, liver, lymph node, comprehensive genomic profiling

## Abstract

Attempts at performing endoscopic ultrasound-guided tissue acquisition (EUS-TA) with a 19G needle are increasing because histological diagnosis and comprehensive genomic profiling are a necessity. However, the diagnostic ability of the 19G fine-needle biopsy (FNB) needle, especially the third-generation FNB needle, is unclear and has been retrospectively reviewed. The 19G TopGain needle was used in 147 patients and 160 lesions between September 2020 and December 2021. The technical success rate of the biopsies was 99.4% (159/160). The early adverse event rate was 4.1% (6/147), and moderate or severe adverse event rate occurrence was 2.0% (3/147). The sensitivity, specificity, and accuracy of the 19G TopGain needle for 157 lesions with a confirmed diagnosis were 96.7%, 100%, and 96.8%, respectively. Rescue EUS-TA using the 19G TopGain needle was performed for nine lesions, and a successful diagnosis was made in six of these lesions (66.7%). The diagnostic ability of EUS-TA using the third-generation 19G TopGain needle was favorable. However, the use of 19G FNB needles may increase adverse events. Therefore, EUS-TA with a 19G FNB needle is mainly indicated in lesions where comprehensive genomic profiling may be necessary or the diagnosis could not be determined via EUS-TA using the 22G needle.

## 1. Introduction

Endoscopic ultrasound-guided tissue acquisition (EUS-TA) is used to diagnose abdominal tissue lesions in organs such as the pancreas, liver, and lymph node tissue. Although EUS-TA has traditionally been performed to differentiate between benign and malignant lesions via a cytological diagnosis, the quality and quantity of specimens have recently been expected to be sufficient for histological diagnosis and genomic testing. Therefore, improved needles are being developed; however, it is necessary to develop needles capable of obtaining better specimens [1]. 

The usefulness of a fine-needle biopsy (FNB) needle, which can obtain more optimal specimens for histological diagnosis than the conventional fine-needle aspiration (FNA) needle, has been widely reported [2,3,4,5,6,7,8,9,10]. In a meta-analysis of 51 studies including 5330 lesions, the rates of diagnostic accuracy, technical success, and adverse events using the FNB needle were reported as 90.82%, 99.71%, and 0.59%, respectively [11]. 

The Tru-cut biopsy needle (Quick-Core, Cook Medical, Limerick, Ireland) is considered a first-generation FNB needle. However, it does not provide superior diagnostic accuracy or technical success as compared to the FNA needle due to the technical difficulty of its use [12,13,14]. The reverse bevel needle (ProCore, Cook Medical, Limerick, Ireland), a second-generation FNB needle, has also demonstrated comparable diagnostic performance to either the FNA needle or the first-generation FNB needle [15,16,17], except in two prospective studies [8,18]. The Franseen needle (Acquire, Boston Scientific Corporation, Massachusetts, USA; SonoTip TopGain, Medi-Globe, Achenmuhle, Germany), fork-tip needle (SharkCore, Medtronic Corporation, Newton, MA), and the forward-facing bevel needle (20G EchoTip ProCore, Cook Medical, Limerick, Ireland) are novel, third-generation needles [19]. Prospective studies comparing the third-generation FNB needles to FNA needles and first- or second-generation FNB needles have reported that the third-generation FNB needles yield superior specimen volume and diagnostic performances [7,9,10,20]. Therefore, third-generation FNB needles are considered the first choice for EUS-TA. A direct comparison of the third-generation FNB needles revealed no differences in diagnostic performances between the needles [21,22,23,24].

Clinical data regarding the 19G FNB needle are inadequate [25,26,27,28], as the 20G or 22G FNB needles are used more commonly. Iwashita et al. reported that the use of a 19G second-generation FNB needle instead of a 19G FNA needle improved diagnostic accuracy (90.0% vs. 79.1%; *p* = 0.039) [26]. DeWitt et al. reported a higher diagnostic accuracy (88% vs. 62%; *p* = 0.001) and a longer median specimen length (19.4 mm vs. 4.3 mm; *p* = 0.001) from a 19G second-generation FNB needle than from a 19G first-generation FNB needle [28]. In contrast, only one study has reported the utility of 19G third-generation FNB needles [25]. Takahashi et al. performed a randomized comparative study involving 30 patients with pancreatic cancer who underwent biopsy with either a 19G third-generation FNB needle, 19G FNA needle, or 22G third-generation FNB needle, and found that the specimens obtained using a 19G FNB needle had significantly greater tissue area than specimens obtained using the other needle types [25].

Additionally, comprehensive genomic profiling (CGP) requires a large volume of specimens. The larger the diameter of the needle, the higher the success rate of the analysis. Therefore, expectations from the 19G FNB needle are increasing [29,30,31]. The technical success rate of the 19G needle is lower than that of the 22G needle due to the technical difficulty of the use of the 19G needle [32,33]. Clinical data regarding the use of the 19G FNB needle remain unclear, especially for the third-generation FNB needle.

TopGain, a third-generation Franseen FNB needle, is made of stainless steel and has more flexibility than other third-generation FNB needles made of nitinol or cobalt chrome for the purpose of widening the penetration angle of the needle (Figure 1). Here, the clinical outcomes of patients who underwent EUS-TA using a 19G TopGain needle at our institution were retrospectively reviewed.

## 2. Materials and Methods

### 2.1. Study Design and Ethics

This study was a single-arm, single-center, retrospective study. The study was approved by National Cancer Center Institutional Review Board (approval number: 2018-149). 

### 2.2. Patients and Data Collection

This study included all patients who underwent EUS-TA with a 19G TopGain needle at our institution between September 2020 and December 2021. Patient age, sex, target organ, target lesion diameter, puncture site, number of punctures, and final clinical diagnosis were extracted from the medical records.

The primary study endpoint was the diagnostic ability (sensitivity, specificity, and accuracy) for benign and malignant lesions. The secondary endpoints included the technical success rate, adverse event rate, and CGP analysis success rate. 

### 2.3. EUS-TA Procedure

A linear echo-endoscope (GF-UCT260; Olympus Optical, Tokyo, Japan) was used in all patients. Suction was performed via the slow-pull method and the number of strokes was approximately 20 in all patients. Rapid on-site evaluation (ROSE) with Diff-Quik staining was performed using a portion of the obtained specimens. Several other methods, such as using an alternative type or gauge of the needle, changing the puncture route, and changing the operator, were attempted when malignant cells were not observed on ROSE, but the target lesion was suspected to be malignant prior to the procedure. The procedure was completed with a maximum of six punctures per lesion.

The 19G TopGain needle is the first choice for unresectable lesions and patients who may undergo CGP analysis at our institution today. However, when we first started using 19G TopGain, we used it for a variety of cases. Therefore, non-unresectable lesions were also included in this study. This needle is also used as a rescue needle in patients in whom the lesions are suspected to be malignant, but no malignant cells have been observed in the ROSEs of specimens obtained using 22G needles. The 22G FNB needle is used for non-unresectable lesions, such as resectable or borderline resectable pancreatic cancer, while the 22G or 25G FNA needles are used for lesions that are difficult or expected to be difficult to puncture with a 19G or 22G FNB needle due to the location or size of the lesion. 

The OncoGuide^TM^ NCC Oncopanel system (NOP; Sysmex Corporation, Hyogo, Japan) was used when it was clinically determined that the patient required CGP analysis.

### 2.4. Histological Evaluation

After ROSE, the remaining specimen was preserved in 10% formalin for subsequent histological diagnosis. Additionally, a cytological diagnosis was made using the puncture needle washing solution and the ROSE preparation. Histological and cytological diagnoses were performed by two pathologists, and a definite diagnosis of malignancy was made only when the histological diagnosis was malignant and/or the cytological diagnosis was class 4 or 5 according to the Papanicolaou classification system.

### 2.5. Study Definitions

In patients in whom two or more types of needles were used in a single lesion, only specimens obtained using 19G TopGain needles were analyzed in this study. 

Histological diagnosis was defined as malignant if it was adenocarcinoma, adenosquamous carcinoma, acinar cell carcinoma, intraductal papillary mucinous carcinoma, neuroendocrine tumor, neuroendocrine carcinoma, mixed neuroendocrine non-neuroendocrine neoplasms, cholangiocellular carcinoma, carcinoma, or malignant lymphoma. Malignancy was clinically defined when the clinical course was consistent with a histological diagnosis of malignancy or when malignancy was confirmed in the postoperative specimen. In contrast, lesions were defined as benign when the postoperative specimen was confirmed as benign, or any malignancy was determined to be negative after a follow-up of at least six months. Lesions that could not be definitively diagnosed as benign or malignant according to the above definitions were considered indeterminate and were excluded from the analysis for diagnostic ability. Auto-immune pancreatitis was diagnosed according to the Japanese Diagnostic Criteria [34]. Desmoid fibrosis was considered a benign disease. 

The biopsy was considered technically successful when it was possible to puncture the target lesion and obtain a specimen using EUS-TA, and technically unsuccessful when it was difficult to puncture the target lesion with a 19G TopGain needle or obtain no specimens. 

Adverse events were evaluated only in cases where 19G TopGain alone was used according to the classification developed by the American Society for Gastrointestinal Endoscopy workshop, and were divided into intraoperative adverse events, early adverse events (up to 14 days), and late adverse events (after 14 days) [35].

### 2.6. Statistical Analysis

Data were analyzed using SPSS version 27.0 (IBM, Armonk, NY, USA) statistical software. Continuous variables are expressed as median (range) and categorical variables as numbers (percentages). The diagnostic ability of the needles, technical success rate, and adverse event rate were analyzed using proportions and 95% confidence intervals (CI).

## 3. Results

### 3.1. Patient Background Characteristics

Of the 564 EUS-TA procedures and 678 lesions performed at our institution between September 2020 and December 2021, a 19G TopGain needle was used in 147 patients (26.0%) and 160 lesions (24.7%) (Table 1). The median patient age was 70 years (range: 15–92 years), and 55.1% of the patients were men. The lesions included pancreatic lesions (76.3%; 122/160), hepatic lesions (11.3%; 18/160), lymph node lesions (10.6%; 17/160), intra-abdominal masses (1.3%; 2/160), and duodenal submucosal tumors (0.6%; 1/160), with a median lesion size of 30 mm (range: 7.4–100 mm). Of the 122 pancreatic lesions, 33.6% (41/122) were in the pancreatic head and 66.4% (81/122) were in the pancreatic tail. The diagnosis prior to EUS-TA was a suspected malignant lesion in 89.4% (143/160) of the patients, a benign lesion in 1.9% (3/160) of the patients, and 8.8% (14/160) of the patients had an uncertain diagnosis. 

### 3.2. Procedure Outcomes

The 19G TopGain needle was used as the first needle in 92.5% (148/160) of lesions and as the second needle in 7.5% (12/160) of lesions (Table 2). Among these 12 lesions, the 22G SharkCore (Medtronic, Tokyo, Japan) needle was used as the first needle in 11 lesions and the 22G TopGain needle was used as the first needle in 1 lesion. A total of 118 lesions (73.8%) were punctured from the stomach, 38 (23.8%) from the duodenum, 3 (1.9%) from both the stomach and duodenum, and 1 (0.6%) from the jejunum. The median number of punctures per lesion was 3 (range: 0–6).

### 3.3. Final Diagnosis

The final diagnosis was a malignant lesion in 150 lesions (93.8%), a benign lesion in 7 lesions (4.4%), and undetermined in 3 lesions (1.9%) (Table 3). The sensitivity, specificity, positive predictive value, negative predictive value, and accuracy of the 19G TopGain needle for 157 lesions with a confirmed diagnosis were 96.7% (145/150) (95%CI, 92.2–98.8%), 100% (7/7) (95%CI, 59.6–100%), 100% (145/145) (95%CI, 96.9–100%), 58.3% (7/12) (95%CI, 31.9–80.7%), and 96.8% (152/157) (95%CI, 92.6–98.8%), respectively (Figure 2 and Table 4).

The five cases of false negatives with the 19G needle were pancreatic cancer in four cases and NET in one case; one case was diagnosed after retesting with 19G, one case was diagnosed after retesting with 22G, one case was diagnosed after percutaneous biopsy for liver metastases, and two cases were not retested but were clinically diagnosed malignant lesions.

### 3.4. Clinical Outcomes

The technical success rate of the biopsies was 99.4% (159/160) (95%CI, 96.2–100%) (Table 5). Of the 148 lesions for which the 19G TopGain needle was selected as the first needle, 146 (98.6%) were completed using the 19G TopGain needle alone and 2 (1.4%) required a different needle type due to technical failure or failure to achieve the expected diagnosis via ROSE. One lesion was difficult to puncture using the 19G TopGain and required the use of a 22G FNB needle. Puncture in the other lesion was technically successful; however, no malignant cells were observed via ROSE. Although additional acquisition using a 22G FNA needle was performed, no malignant findings were finally obtained on histological diagnosis. The TopGain needle served as a rescue needle for nine lesions where the ROSE results of the specimen obtained with the first needle were unexpected, or an adequate specimen could not be obtained using the first needle. The ROSE results of the specimen obtained with the 19G TopGain needle were consistent with the preoperative diagnosis in 66.7% (6/9) of these patients (Figure 3 and Table 6). Histological diagnosis was obtained for all three lesions where the expected diagnosis was not obtained via ROSE.

Of the 150 lesions diagnosed as malignant in the final pathology in this study, 24 were analyzed using NOP, as they were clinically deemed to require CGP and met the criteria for NOP analysis. All 24 (100%) lesions were successfully analyzed using NOP.

Of the two patients with bleeding, one improved with only conservative treatment, and one required endoscopic hemostasis. Both patients with intra-abdominal infections required hospitalization and endoscopic drainage. The patient with pancreatitis improved with three days of conservative treatment. The patient with aspiration pneumonia was treated with antibiotics for three days and improved.

19G TopGain was useful as a rescue needle in cases 1, 2, 4, 5, 8, and 9. Case 5 was followed up for 1 year after EUS-TA, during which the pancreatic lesion reduced naturally, after which mass-forming pancreatitis was suspected. Case 7 was followed up for 10 months after EUS-TA during which the pancreatic lesion did not change, after which the lesion was not diagnosed, and the follow-up continued. The ROSE of case 8 was class 2 in the specimens obtained from both 22G FNB and 19G TopGain. However, the ROSE showed stromal cells in only one of the specimens.

EUS-TA, ultrasound-guided tissue acquisition; FNB, fine-needle biopsy; NET, neuroendocrine tumor; PDAC, pancreatic ductal adenocarcinoma; ROSE, rapid on-site evaluation; Panc, pancreas 

### 3.5. Adverse Events

The early adverse event rate was 4.1% (6/146) (95%CI, 1.70–8.87%), and the rate of moderate or severe adverse events was 2.1% (3/146) (95%CI, 0.43–6.14%) (Table 5). Two patients experienced bleeding, including one who required endoscopic hemostasis. Two patients experienced intra-abdominal infection and both required EUS-guided abscess drainage. One experienced pancreatitis, and one experienced aspiration pneumonia after the procedure. Both patients, managed conservatively, improved within three days. No intraoperative or late adverse events were observed.

## 4. Discussion

The clinical outcomes of EUS-TA using only 19G TopGain, the third-generation FNB needle, are reported in this study. To the best of our knowledge, this is the first report regarding the diagnostic ability of the third-generation 19G FNB needle. The diagnostic accuracy of EUS-TA using the 19G TopGain needle was 96.8%. A previous meta-analysis of 51 studies including 5330 lesions reported a 90.8% diagnostic accuracy of EUS-TA using the FNB needle [11]. Most of the studies in the meta-analysis included 22G or 25G needles. Previous reports on the clinical outcomes of EUS-TA using 19G needles are limited and report the diagnostic accuracy as 62–95.5% (Table 7) [26,28,32,33,36,37,38,39,40]. These studies imply that the diagnostic accuracy of the 19G FNB needle is 62–92.2%; however, these findings are limited to the use of first- and second-generation 19G FNB needles [26,28,37]. The diagnostic accuracy of EUS-TA using the third-generation 19G FNB needle here was not inferior to the diagnostic accuracy reported in those studies. 

The 19G needle is reported to be less maneuverable due to its larger diameter, resulting in a lower technical success rate. Laqulere et al. reported a significantly higher technical success rate when a 22G nitinol FNA needle was used as compared to when a 19G nitinol FNA needle was used [33]. However, the 19G TopGain needle used in this study is expected to have better maneuverability and improved puncture performance due to the flexibility of its stainless-steel construction and its sharp needle tip, respectively. The rate of technical success in this study was 99.4%. However, technical success is also dependent on the site of the puncture, target of the puncture, diameter of the target, and target disease. Therefore, a randomized comparison study of the third-generation 19G and 22G FNB needles is required to evaluate the puncture performance of the 19G TopGain needle. 

As specimens collected using the 19G FNB needle are large, the expected rate of successful analysis at CGP is high [31,41] (Figure 4). In this study, all 24 patients who clinically required CGP and met the criteria for NOP analysis in the precheck by pathologists were successfully analyzed. In a previous multivariate analysis, Ikeda et al. reported that the use of FNB needles (compared to FNA needles) and 19G needles (compared to 22G needles) were significantly associated with a higher rate of specimens that met the criteria for NOP analysis [31]. Hisada et al. reported that the rate of the specimen that meets the analysis criteria for NOP is high when the 19G TopGain needle is used to perform EUS-TA for suspected pancreatic cancer, which is consistent with the results of the current study [41]. Therefore, the 19G third-generation FNB needle is useful for EUS-TA that may require CGP.

The usefulness of the 19G FNB needle as a rescue needle was also analyzed in this study. Rescue EUS-TA using the 19G TopGain needle was performed for nine lesions in which diagnosis could not be confirmed via ROSE after EUS-TA using a 22G FNB needle. Diagnosis was confirmed via ROSE after EUS-TA using the 19G TopGain needle in six of these lesions (66.7%), and the histological diagnosis was consistent with the final clinical diagnosis in all nine lesions, suggesting the usefulness of the 19G TopGain needle as a rescue needle. In low-cellularity tumors, such as pancreatic cancer, only interstitial components may be collected, and the area of the tumor in the specimen may be insufficient, even when the target lesion has been successfully punctured. In lesions where the diagnosis cannot be made or the specimen volume is insufficient when the 22G FNB needle is used, the 19G FNB needle is expected to enable a pathological diagnosis due to the large amount of tissue collected. In facilities that do not routinely perform ROSE, the 19G FNB needle may be useful for re-testing lesions that were not diagnosed using the 22G FNB needle.

The overall rate of adverse events in this study was 4.1% (6/146), and the rate of moderate or more severe adverse events was 2.1% (3/146). One bleeding case and two abdominal infection cases required endoscopic invasive treatment. These adverse events could be caused by the large diameter of the 19G FNB needle. A previous, multicenter, retrospective study reported 234 adverse events (1.7%) in 13,566 patients undergoing EUS-TA using various needle sizes and types [42]. However, the rates of adverse events between needles of different sizes were not compared. Li et al. reported that EUS-TA using 25G needles tended to result in fewer adverse events than EUS-TA using 22G, 20G, or 19G needles, although the differences were not significant [11]. The adverse event rates have been reported as 0–16% in previous reports of EUS-TA using a 19G needle [26,32,33,39,40], though this increased rate may simply be due to the size of the 19G needle. 

This study has some limitations, one of which is the possibility of selection bias due to the single-center, retrospective, single-arm study design. However, there are no previous reports regarding the diagnostic performance of the third-generation 19G FNB needle, the examination of which would be valuable. 

In conclusion, the diagnostic accuracy and the technical success rate of EUS-TA using the third-generation 19G TopGain needle were favorable. However, the use of 19G FNB needles may increase the incidence of adverse events. Therefore, EUS-TA with a 19G FNB needle is mainly indicated in lesions where CGP may be necessary, such as advanced unresectable cancer, or in lesions where the diagnosis was not determined via EUS-TA using a 22G needle.

## Figures and Tables

**Figure 1 diagnostics-13-00450-f001:**
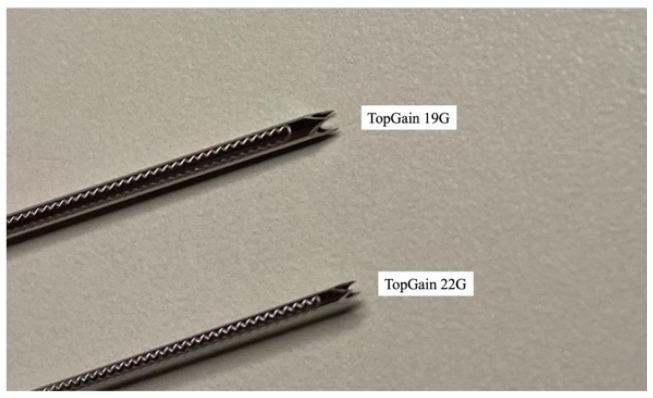
TopGain, 19G, and 22G. TopGain, a third-generation, Franseen, fine-needle biopsy (FNB) needle, is made of stainless steel, which is more flexible than other third-generation FNB needles made of nitinol or cobalt chrome. The 19G TopGain needle is expected to be as easy to use as a 22G FNB needle.

**Figure 2 diagnostics-13-00450-f002:**
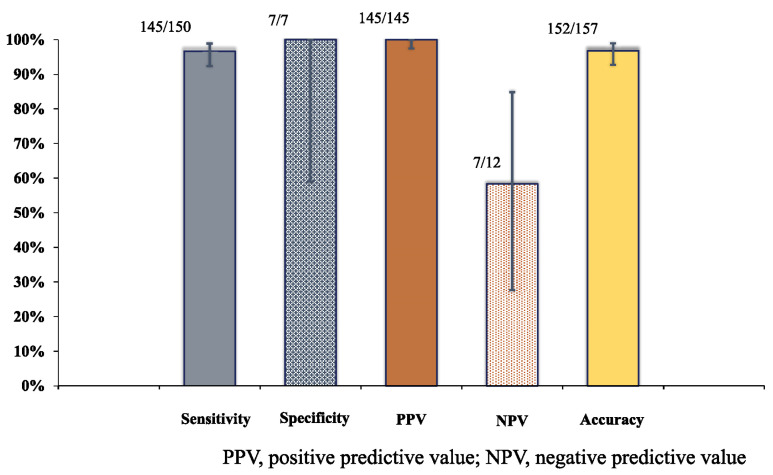
The diagnostic ability of the 19G TopGain needle. The error bar showed 95% confidence interval (95%CI). The sensitivity, specificity, positive predictive value, negative predictive value, and accuracy of the 19G TopGain needle for 157 lesions with a confirmed diagnosis were 96.7% (145/150) (95%CI, 92.2–98.8%), 100% (7/7) (95%CI, 59.6–100%), 100% (145/145) (95%CI, 96.9–100%), 58.3% (7/12) (95%CI, 31.9–80.7%), and 96.8% (152/157) (95%CI, 92.6–98.8%), respectively.

**Figure 3 diagnostics-13-00450-f003:**
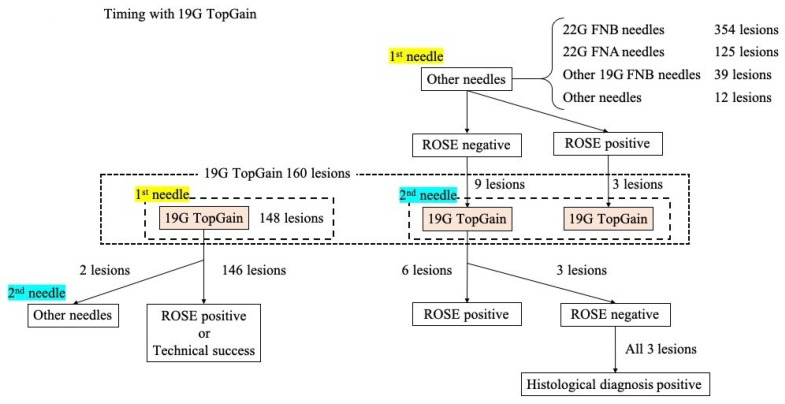
The clinical use of a 19G TopGain needle. A 19G TopGain needle was selected as the first needle in 92.5% (148/160) of lesions and as the second needle in 7.5% (12/160) of lesions. The TopGain needle served as a rescue needle for nine lesions where the rapid on-site evaluation (ROSE) results of the specimen obtained with the first needle of choice were unexpected, or an adequate specimen could not be obtained using the first needle. The ROSE results of the specimen obtained with the 19G TopGain needle were consistent with the preoperative diagnosis in 66.7% (6/9) of these patients. All three lesions in which the expected diagnosis was not obtained in the ROSE of specimens obtained with either the first or second needle were diagnosed histologically.

**Figure 4 diagnostics-13-00450-f004:**
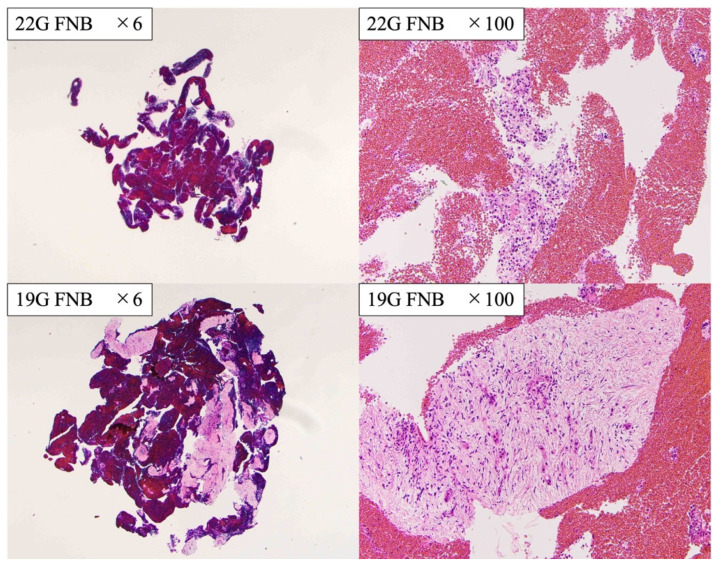
The comparison of histological findings between 19G FNB needle and 22G FNB needle.

**Table 1 diagnostics-13-00450-t001:** Characteristics of the patients.

Patient Characteristics	N = 147 Cases (160 Lesions)	
Median age, year (range)	70	(15–92)
Sex, male (%)	81/147	(55.1%)
Objects (%)	N = 160 lesions	
Pancreas	122/160	(76.3%)
Liver	18/160	(11.3%)
Lymph node	17/160	(10.6%)
Others	3/160	(1.9%)
Median diameter of the object, mm (range)	30	(7.4–100)
The lesion of the pancreas (%)	N = 122 lesions	
Head	41/122	(33.6%)
Body or tail	81/122	(66.4%)
Preprocedural diagnosis (%)	N = 160 lesions	
Malignancy	143/160	(89.4%)
Benign	3/160	(1.9%)
Indeterminate	14/160	(8.8%)

**Table 2 diagnostics-13-00450-t002:** Procedure details.

Procedure Details	160 Lesions	
19G TopGain as the first needle (%)	148	(92.5%)
19G TopGain as the second needle (%)	12	(7.5%)
First needle: 22G SharkCore (%)	11/12	(91.7%)
First needle: 22G TopGain (%)	1/12	(8.3%)
Puncture site (%)		
Stomach	118	(73.8%)
Duodenum	38	(23.8%)
Stomach and duodenum	3	(1.9%)
Jejunum	1	(0.6%)
Median number of punctures per lesion (range)	3	(0–6)
Suction, slow pull (%)	160	(100%)
Rapid on-site evaluation (%)	160	(100%)

**Table 3 diagnostics-13-00450-t003:** Clinical and histological diagnosis.

Clinical and Histological Diagnosis	160 Lesions	
Malignant lesion (%)	150	(93.8%)
Benign lesion (%)	7	(4.4%)
Indeterminate lesion (%)	3	(1.9%)
Target organ (%)		
**Pancreas**	**122/160**	**(76.3%)**
Adenocarcinoma	96	(78.7%)
Adenosquamous carcinoma	6	(4.9%)
Neuroendocrine tumor	6	(4.9%)
Neuroendocrine carcinoma	4	(3.3%)
Intraductal papillary mucinous carcinoma	1	(0.8%)
Acinar cell carcinoma	1	(0.8%)
Malignant lymphoma	1	(0.8%)
Autoimmune pancreatitis	4	(3.3%)
Normal pancreatic tissue	1	(0.8%)
Indeterminate lesion	2	(1.6%)
**Liver**	**18/160**	**(11.3%)**
Pancreatic ductal adenocarcinoma	6	(33.3%)
Cholangiocellular carcinoma	5	(27.8%)
Pancreatic adenosquamous carcinoma	2	(11.1%)
Mixed neuroendocrine non-neuroendocrine neoplasm of the pancreas	2	(11.1%)
Carcinoma of the bile duct	1	(5.6%)
Neuroendocrine tumor of the pancreas	1	(5.6%)
Indeterminate lesion	1	(5.6%)
**Lymph node**	**17/160**	**(10.6%)**
Malignant lymphoma	4	(23.5%)
Cholangiocellular carcinoma	3	(17.6%)
Pancreatic ductal adenocarcinoma	3	(17.6%)
Neuroendocrine carcinoma of the pancreas	2	(11.8%)
Carcinoma of the bile duct	1	(5.9%)
Neuroendocrine tumor of the pancreas	1	(5.9%)
Mixed neuroendocrine non-neuroendocrine neoplasm of the pancreas	1	(5.9%)
Carcinoma of the gallbladder	1	(5.9%)
Small cell lung carcinoma	1	(5.9%)
**Abdominal mass**	**2/160**	**(1.3%)**
Pancreatic adenosquamous carcinoma	1	(50%)
Desmoid fibrosis	1	(50%)
**Submucosal tumor**	**1/160**	**(0.6%)**
Brunner’s glands	1	(100%)

**Table 4 diagnostics-13-00450-t004:** The diagnostic ability of a 19G TopGain needle.

		Final Diagnosis		Total
		Malignancy	Benign	
EUS-FNA outcome	Malignancy	145	0	145
	Benign	5	7	12
Total		150	7	157

**Table 5 diagnostics-13-00450-t005:** Clinical outcomes.

Clinical Outcomes	N = 147 Cases (160 Lesions)	
Technical success (%)	159/160	(99.4%)
Technical success with only TopGain19G(%)	146/148	(98.6%)
Adverse events during the procedure (%)	0/146	(0%)
Early adverse events (%)	6/146	(4.1%)
Bleeding	2/6	(33.3%)
Infection	2/6	(33.3%)
Pancreatitis	1/6	(16.7%)
Aspiration pneumonia	1/6	(16.7%)
Late adverse events (%)	0/146	(0%)
Success of CGP analysis	24/24	(100%)

**Table 6 diagnostics-13-00450-t006:** Details of nine cases in which 19G TopGain was used as a rescue needle.

Case	Object	Diameter (mm)	Preprocedural Diagnosis	Puncture Site	Puncture by the First Needle			Rescue Puncture by TopGain19G		Final Diagnosis
					The Type of the First Needle	The Number of Punctures	ROSE (Papanicolaou Classification)	The Number of Punctures	ROSE (Papanicolaou Classification)	
1	Panc head	24.7	PDAC	duodenum	FNB 22G	3	1	2	5	PDAC
2	Panc body~tail	7.4	NET	stomach	FNB 22G	3	1	1	3	NET
3	Panc body~tail	24.6	PDAC	stomach	FNB 22G	4	1	1	1	PDAC
4	Panc body~tail	8.9	NET	stomach	FNB 22G	3	inadequate	3	3	NET
5	Panc body~tail	17.6	NET	stomach	FNB 22G	2	inadequate	1	3	No tumor seen
6	Panc body~tail	13.5	PDAC	stomach	FNB 22G	4	3	1	3	PDAC
7	Panc body~tail	8.1	NET	stomach	FNB 22G	4	1	1	1	Benign
8	Abdominal mass	24.4	Not diagnosed	duodenum	FNB 22G	3	2	2	2 (stromal cell)	Desmoid fibrosis
9	Panc head	20	PDAC	stomach	FNB 22G	5	1	2	5	PDAC

**Table 7 diagnostics-13-00450-t007:** Review of key literature regarding the EUS-tissue acquisition using 19G needle.

Literatures	Design	Cases	Type of Needle	Material Used in the Needle	Diagnostic Accuracy	Technical Success	Adverse Events
[36]	Prospective randomized	60	FNA/19G (ECHO19, COOK)	stainless	86.7%	91.7%	0%
		57	FNA/22G (ECHO22, COOK)		78.9%	100%	4.6%
[37]	Prospective, single-arm	114	FNB/19G (Procore, COOK)	stainless	92.9%	98.2%	0%
[28]	Prospective randomized	44	FNB/19G (Procore, COOK)	stainless	88%	94%	13%
		41	FNB/19G (Quick-Core, COOK)	stainless	62%	86%	16%
[38]	Prospective, single-arm	111	FNA/19G (Echotip, COOK)	stainless	95.5%	99.1%	0%
[32]	Prospective, single-arm	246	FNA/19G (Expect19 Flex, Boston)	Nitinol	73.6%	92.7%	2.4%
[26]	Prospective randomized	55	FNB/19G (Procore, COOK)	stainless	90%	98.2%	3.6%
		55	FNA/19G (Echotip, COOK)	stainless	79.1%	97.3%	
[39]	Prospective randomized, crossover	46	FNA/19G (EZ shot3 plus, Olympus)	nitinol	68%	100%	4.3%
			FNA/19G (EZ shot2, Olympus)	stainless	66%	93.4%	
[33]	Prospective randomized	59	FNA/19G (Expect19 Flex, Boston)	nitinol	69.5%	86.4%	15%
		63	FNA/22G (various)	various	87.3%	100%	8%
[40]	Retrospective, single-arm	88	FNA/19G (Expect slimline flex Boston)	Nitinol	92.9%	100%	0%
		113	FNA/19G (Expect, Boston)	Cobalt chromium	94.4%		
Present study	Retrospective, single-arm	160	FNB/19G (TopGain, Medico’s HIRATA)	stainless	96.8%	99.4%	4.1%

EUS, ultrasound-guided; FNA, fine-needle aspiration; FNB, fine-needle biopsy.

## Data Availability

The data presented in this study are available on request from the corresponding author.

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
