# Peer review of "Diagnostic Ability of Endoscopic Ultrasound-Guided Tissue Acquisition Using 19-Gauge Fine-Needle Biopsy Needle for Abdominal Lesions"

_diagnostics, 2023, doi:10.3390/diagnostics13030450_

Round 1

Reviewer 1 Report

1.      147 patients and 160 lesions among 564 EUS-TA procedures and 678 lesions in this report received EUS-TA with 19G needle as the first procedure chiefly for the unresectable lesions. Does this imply that this 19G needle is more suitable for advanced lesions?

2.      The 19G biopsy needle was suspected to be with higher adverse events, what were the causes of these events? More detailed description of these adverse events may be considered.

3.      How about the maneuverability of this new 19G needle in real world practice as compare with other 19G needles or smaller caliber needles?

4.      In the outcome section: The TopGain needle served as a rescue needle for nine lesions. The ROSE results of the specimen obtained were considered as positive for malignancy in 66.7% (6/9). However, in Table 6 showed that only two were class 5 and the other 7 cases were class 1~3. (The cytological diagnosis of malignancy was defined when class 4 or 5 according to the Papanicolaou classification system was noted.)

Author Response

RESPONSE TO REVIEWER 1

We wish to thank the reviewer for the detailed review of our manuscript and providing insightful comments for improving it. We have responded to the comments and highlighted the changes made in the revised manuscripts based on each comment.

  1. 147 patients and 160 lesions among 564 EUS-TA procedures and 678 lesions in this report received EUS-TA with 19G needle as the first procedure chiefly for the unresectable lesions. Does this imply that this 19G needle is more suitable for advanced lesions?

    Response: Thank you for the pertinent comment. EUS-TA using the 19G FNB is expected to yield a larger specimen volume, so we believe it is the first choice for cases that may require comprehensive genomic profiling in the future. It is also a good choice for cases where other puncture needles, such as the 22G needle, have not been able to make a diagnosis due to insufficient specimen volume. On the other hand, the possibility of increased adverse events due to the larger needle diameter cannot be denied, so we do not think it should be the first choice for resectable lesions. We have included the following information in the last paragraph(line 340) of the discussion.

    Original: Therefore, EUS-TA with a 19G FNB needle is mainly indicated in lesions where CGP may be necessary or in lesions where the diagnosis was not determined via EUS-TA using a 22G needle.
    Revised: Therefore, EUS-TA with a 19G FNB needle is mainly indicated in lesions where CGP may be necessary, such as advanced unresectable cancer, or in lesions where the diagnosis was not determined via EUS-TA using a 22G needle.

  2. The 19G biopsy needle was suspected to be with higher adverse events, what were the causes of these events? More detailed description of these adverse events may be considered.

    Response: Thank you for your valuable comment. The adverse events that required additional procedures in this study were bleeding in one case and intra-abdominal infection in two cases. In one case of bleeding, hemostasis was achieved by endoscopic hemostasis. Both cases of intra-abdominal infection were successfully treated with endoscopic drainage. All other adverse events were treated conservatively. As for the two cases of abscess, one was a case of auto-immune pancreatitis (AIP) and the other was a case of pancreatic cancer; the AIP case was not identified at the time of EUS, but may have been caused by leakage of pancreatic juice due to injury to the pancreatic duct. In the case of pancreatic cancer, necrotic tissue may have spread into the abdominal cavity. In any case, we cannot deny the possibility that all three cases were caused by the large puncture needle. Therefore, the section 3.5 on adverse events (line 261) has been revised as follows.

    Original: Two patients experienced bleeding, including one who required endoscopic hemostasis. Two patients experienced intra-abdominal infection, one experienced pancreatitis, and one experienced aspiration pneumonia after the procedure. No intraoperative or late adverse events were observed.
    Revised: Two patients experienced bleeding, including one who required endoscopic hemostasis. Two patients experienced intra-abdominal infection and both required EUS-guided abscess drainage. One experienced pancreatitis, and one experienced aspiration pneumonia after the procedure. Both patients managed conservatively improved within three days. No intraoperative or late adverse events were observed.

    Discussion (line 323) has been revised as follows.
    Original: The overall rate of adverse events in this study was 4.1% (6/146), and the rate of moderate or more severe adverse events was 2.1% (3/146).
    Revised: The overall rate of adverse events in this study was 4.1% (6/146), and the rate of moderate or more severe adverse events was 2.1% (3/146). One bleeding case and two abdominal infection cases required endoscopic invasive treatment. These adverse events could be caused by the large diameter of 19G FNB needle.

  3. How about the maneuverability of this new 19G needle in real world practice as compare with other 19G needles or smaller caliber needles
    Response: Thank you for your important question. Due to the characteristics of the needle, that is made of stainless steel and has more flexibility than other third-generation FNB needles made of nitinol or cobalt chrome, we believe that it is more flexible and easier to operate than other 19G needles. In particular, it has relatively high operability even when puncturing sites that are considered difficult to puncture with a 19G needle, such as duodenum. We believe that this operability is the reason for the high technical success rate in this study. We added it in Introduction (line 82) as followed.

    Original: TopGain, a third-generation Franseen FNB needle, is made of stainless steel and has more flexibility than other third-generation FNB needles made of nitinol or cobalt chrome (Figure 1).
    Revised: TopGain, a third-generation Franseen FNB needle, is made of stainless steel and has more flexibility than other third-generation FNB needles made of nitinol or cobalt chrome, for the purpose of widening the penetration angle of the needle (Figure 1).

  4. In the outcome section: The TopGain needle served as a rescue needle for nine lesions. The ROSE results of the specimen obtained were considered as positive for malignancy in 66.7% (6/9). However, in Table 6 showed that only two were class 5 and the other 7 cases were class 1~3. (The cytological diagnosis of malignancy was defined when class 4 or 5 according to the Papanicolaou classification system was noted.)

    Response: Thank you for pointing out a very important point: the 66.7% (6/9) rate of ROSE diagnosis at 19G rescue after 22G puncture is due to "the fact that 22G did not produce the preoperative expected ROSE cytology result, but 19G did produce the expected ROSE cytology result". Malignant lesions in this study include not only those classified 4 or 5 in the Papanicolau classification, but also those diagnosed as NETs by histological diagnosis. For example, in the case of NETs, findings suggestive of NETs by ROSE are often seen even in class 3 of the Papanicolau classification. In addition, in cases where a 22G puncture specimen showed only blood and gastrointestinal wall contamination, if the ROSE after a 19G puncture showed pancreatic acinar cells, the specimen was considered to have been rescued by the 19G puncture. Therefore, the Papanicolaou classification and the result of rescue are different. Since there was a discrepancy in the definitions, we have made the following correction in the Discussion (line 309).

Original: Rescue EUS-TA using the 19G TopGain needle was performed for nine lesions in which malignant cells could not be confirmed via ROSE after EUS-TA using a 22G FNB needle. Malignant cells were confirmed via ROSE after EUS-TA using the 19G TopGain needle in six of these lesions (66.7%), and the histological diagnosis was consistent with the final clinical diagnosis in all lesions, suggesting the usefulness of the 19G TopGain needle as a rescue needle.
Revised: Rescue EUS-TA using the 19G TopGain needle was performed for nine lesions in which diagnosis could not be confirmed via ROSE after EUS-TA using a 22G FNB needle. Diagnosis was confirmed via ROSE after EUS-TA using the 19G TopGain needle in six of these lesions (66.7%), and the histological diagnosis was consistent with the final clinical diagnosis in all 9 lesions, suggesting the usefulness of the 19G TopGain needle as a rescue needle.

Reviewer 2 Report

The authors describe “The 19G TopGain needle is the first choice for unresectable lesions and patients who may undergo CGP analysis at our institution.” but “a 19G TopGain needle was used in 147 patients 159 (26.0%) and 160 lesions (24.7%).”  Why the 19G TopGain needles were not used in the other lesions?  Were they either resectable lesions or benign lesions?

The author should describe how many lesions underwent 1st needle by the other needles in the study period and resulted in ROSE negative and ROSE positive in Figure 3.

The Authors should describe 95% confidence interval of the major endpoints including sensitivity, specificity, accuracy, PPV, NPV, technical success, and the rate of adverse events in the manuscript because there are no comparisons and only point estimates cannot indicate the value of this study appropriately.

The authors should address five lesions which showed false negative by EUS-FNA outcome.

In the discussion, the authors described, “The diagnostic accuracy of EUS-TA using the third-generation 19G FNB needle here was significantly greater than the diagnostic accuracy reported in those studies.” It is saying too much because this study is non-comparative retrospective study and statistical threshold was not prespecified.

The criteria for NOP analysis, as well as brief explanation of CGP, should be described to show the characteristics of the lesions and specimens which were analyzed by CGP and to understand why 19G FNB needle may favor. The authors should also address how many lesions could not meet the criteria.

In table 5, clinical success is 159/160 lesions but technical success with only TopGain19G is 146/148 lesions. Why the failure is greater in the latter group? 

Author Response

RESPONSE TO REVIEWER 2

Thank you for your detailed review of our manuscript and the insightful comments, which helped to improve the manuscript. Please find below a point-by-point response to the comments and the changes made in the revised manuscript based on each comment.

  1. The authors describe “The 19G TopGain needle is the first choice for unresectable lesions and patients who may undergo CGP analysis at our institution.” but “a 19G TopGain needle was used in 147 patients 159 (26.0%) and 160 lesions (24.7%).”  Why the 19G TopGain needles were not used in the other lesions?  Were they either resectable lesions or benign lesions?

    Response: Thank you for pointing this out. When we first started using the 19G FNB needle during this observation period, we mainly used the 22G FNB needle because there was no clear distinction between the two needles and we were not certain about the safety of 19G needle. In the latter observation period, as described in the text, the 19G FNB needle was selected for patients with unresectable lesions and those who may require future genomic profiling or when other puncture needles, such as the 22GFNB needle, failed to make a diagnosis or did not collect sufficient specimens. Because of this, not all cases in this observation period in which the 19G FNB needle was not used are resectable or benign. We have added the following note to section 2.3. EUS-TA procedure (line 111).

    Original: The 19G TopGain needle is the first choice for unresectable lesions and patients who may undergo CGP analysis at our institution.
    Revised: The 19G TopGain needle is the first choice for unresectable lesions and patients who may undergo CGP analysis at our institution today. However, when we first started using 19G TopGain, we used it for a variety of cases. Therefore, non-unresectable lesions were also included in this study.

  1. The author should describe how many lesions underwent 1stneedle by the other needles in the study period and resulted in ROSE negative and ROSE positive in Figure 3.

    Response: Thank you for this helpful suggestion. We are very sorry, but we were able to calculate the ROSE results for the cases using the 19G needle because we knew the ROSE results for all the cases in which ROSE was performed. Since there are many cases in which ROSE was not performed or ROSE results were not recorded for cases in which other needles were used, it is not possible to accurately represent the ROSE results in Figure 3. We have added a breakdown of the frequency of use of other needles in Figure 3.

  1. The Authors should describe 95% confidence interval of the major endpoints including sensitivity, specificity, accuracy, PPV, NPV, technical success, and the rate of adverse events in the manuscript because there are no comparisons and only point estimates cannot indicate the value of this study appropriately.

    Response: Thank you for pointing this out. We agree with the reviewer’s point that the 95% confidence interval would be easier to understand, so we have added the following.

    6 Statistical Analysis (line 156)
    Original: The diagnostic ability of the needles was analyzed using proportions and 95% confidence intervals.
    Revised: The diagnostic ability of the needles, technical success rate, and adverse events rate were analyzed using proportions and 95% confidence intervals (CI).

    3.3 Final Diagnosis (line 184)
    Original: The sensitivity, specificity, positive predictive value, negative predictive value, and accuracy of the 19G TopGain needle for 157 lesions with a confirmed diagnosis were 96.7% (145/150), 100% (7/7), 100% (145/145), 58.3% (7/12), and 96.8% (152/157), respectively (Figure 2 and Table 4).

    Revised: The sensitivity, specificity, positive predictive value, negative predictive value, and accuracy of the 19G TopGain needle for 157 lesions with a confirmed diagnosis were 96.7% (145/150) (95%CI, 92.2-98.8%), 100% (7/7) (95%CI, 59.6-100%), 100% (145/145) (95%CI, 96.9-100%), 58.3% (7/12) (95%CI, 31.9-80.7%), and 96.8% (152/157) (95%CI, 92.6-98.8%), respectively (Figure 2 and Table 4)

    Figure 2
    Original: The sensitivity, specificity, positive predictive value, negative predictive value, and accuracy of the 19G TopGain needle for 157 lesions with a confirmed diagnosis were 96.7% (145/150), 100% (7/7), 100% (145/145), 58.3% (7/12), and 96.8% (152/157), respectively.
    Revised: The sensitivity, specificity, positive predictive value, negative predictive value, and accuracy of the 19G TopGain needle for 157 lesions with a confirmed diagnosis were 96.7% (145/150) (95%CI, 92.2-98.8%), 100% (7/7) (95%CI, 59.6-100%), 100% (145/145) (95%CI, 96.9-100%), 58.3% (7/12) (95%CI, 31.9-80.7%), and 96.8% (152/157) (95%CI, 92.6-98.8%), respectively.

    3.4 Clinical Outcomes (line 207)
    Original: The technical success rate of the biopsies was 99.4% (159/160) (Table 5).
    Revised: The technical success rate of the biopsies was 99.4% (159/160) (95%CI, 96.2-100%) (Table 5).

    3.5 Adverse events (line 260)

    Original: The early adverse event rate was 4.1% (6/147), and the rate of moderate or severe adverse events was 2.0% (3/147) (Table 5).
    Revised: The early adverse event rate was 4.1% (6/147) (95%CI, 1.69-8.81%), and the rate of moderate or severe adverse events was 2.0% (3/147) (95%CI, 0.43-6.10%) (Table 5).

  1. The authors should address five lesions which showed false negative by EUS-FNA outcome.

    Response: Thank you for your comment. One of the 5 cases was punctured twice with a 22G FNB needle and once with a 19G FNB needle; no diagnosis was made, and the patient was retested later with a 19G FNB needle and punctured 4 times, all with a diagnosis of NET. The second case was punctured for the purpose of tissue collection for genomic testing, but the diagnosis was not made, and the patient was switched to standard chemotherapy. The third case was re-punctured at a later date with a 22G FNB needle and a malignant finding was obtained. In the fourth case, puncture to the pancreatic primary tumor yielded only necrotic tissue, and the diagnosis was made by percutaneous biopsy for liver metastases. The fifth case was difficult to puncture with a 19G FNB needle and the diagnosis was not made. Therefore, the needle was changed to 22G, but the diagnosis was still not made. The patient was referred to another hospital without reexamination, but the diagnosis was judged as false negative because the tumor was clinically malignant and consistent. 3.3 The following was added to the Final Diagnosis (line190).

    Revised: The five cases of false-negative with the 19G needle were pancreatic cancer in four cases and NET in one case; one case was diagnosed after retesting with 19G, one case was diagnosed after retesting with 22G, one case was diagnosed after percutaneous biopsy for liver metastases, and two cases were not retested but were clinically diagnosed malignant lesions.

  2. In the discussion, the authors described, “The diagnostic accuracy of EUS-TA using the third-generation 19G FNB needle here was significantly greater than the diagnostic accuracy reported in those studies.” It is saying too much because this study is non-comparative retrospective study and statistical threshold was not prespecified.

    Response: Thank you for pointing this out. As you have pointed out, direct comparison is difficult, and we have corrected the expression in section Discussion (line 280).

    Original: The diagnostic accuracy of EUS-TA using the third-generation 19G FNB needle here was significantly greater than the diagnostic accuracy reported in those studies.
    Revised:  The diagnostic accuracy of EUS-TA using the third-generation 19G FNB needle here was not inferior to the diagnostic accuracy reported in those studies.

  3. The criteria for NOP analysis, as well as brief explanation of CGP, should be described to show the characteristics of the lesions and specimens which were analyzed by CGP and to understand why 19G FNB needle may favor. The authors should also address how many lesions could not meet the criteria.

    Response: Thank you for pointing this out, we have reported our research on CGP in another paper (G Ikeda, et al. DEN 2022: reference number, 31), so we have omitted the details here. We apologize for the inconvenience.

In table 5, clinical success is 159/160 lesions but technical success with only TopGain19G is 146/148 lesions. Why the failure is greater in the latter group? 

Response: Thank you for the insightful comment. In this study, technical success was defined as the ability to puncture the lesion and obtain a specimen, and the rate of technical success for all 160 lesions was 159/160. On the other hand, of the 148 cases in which the 19G FNB needle was used as the first choice, two required a change of puncture needle due to technical difficulty in puncturing or because the ROSE did not produce the expected results. Therefore, puncture was initiated with FNB needle 19G, and the percentage of patients who required a change of puncture needle was 2/148. One of the two cases is considered a technical success because it is a case in which the target was punctured with a 19G FNB needle but the diagnosis was not made by ROSE. The overall technical success is 159/160. We have included the following information in the Clinical Outcomes section (line 207), which we hope you will find useful.

Original: The technical success rate of the biopsies was 99.4% (159/160) (Table 5). Of the 148 lesions for which the 19G TopGain needle was selected as the first needle, 146 (98.6%) were completed using the 19G TopGain needle alone and two (1.4%) required a different needle type due to technical failure or failure to achieve the expected diagnosis via ROSE. One lesion was difficult to puncture and required the use of a 22G FNB needle. The other lesion was technically successful using a 22G FNA needle although no malignant cells were observed via ROSE.
Revised: The technical success rate of the biopsies was 99.4% (159/160) (95%CI, 96.2-100%) (Table 5). Of the 148 lesions for which the 19G TopGain needle was selected as the first needle, 146 (98.6%) were completed using the 19G TopGain needle alone and two (1.4%) required a different needle type due to technical failure or failure to achieve the expected diagnosis via ROSE. One lesion was difficult to puncture by the 19G TopGain and required the use of a 22G FNB needle. Puncture in the other lesion was technically successful; however, no malignant cells were observed via ROSE. Although additional acquisition using a 22G FNA needle was performed, no malignant findings were obtained on histological analysis.

Reviewer 3 Report

Referee of " Diagnostic ability of endoscopic ultrasound-guided tissue acquisition using 19-gauge fine-needle biopsy needle for abdominal lesions "

This is a meaningful and informative article about the usefulness and safety of 19G EUS-FNB for comprehensive genomic profiling.

Generally positive to accept Diagnostic, but some modifications would be desirable.

Minor points

Introduction Line 79-82

The 19G TopGain needle is expected to be as easy to use as the 22G FNB needle (Figure 1).

As the author himself states in the discussion, RCT of the third-generation 19G and 22G FNB needles is required to evaluate the puncture performance of the 19G TopGain needle. Please state only the facts

The 3rd generation Franseen FNB needle, TopGain, is made of stainless steel, which is more flexible than other 3rd generation FNB needles made of nitinol or cobalt chrome, for the purpose of widening the penetration angle of the needle.

Materials and Methods

2.3. EUS-TA Procedure

Line 109  Misprint

suspected to be malignant prior to the procedure..

Line 114-117

The 22G FNB needle is used for resectable lesions, while the 22G or 25G FNA needles are used for lesions that are difficult or expected to be difficult to puncture with a 19G or 22G FNB needle due to the location or size of the lesion.

In case of PDAC, which is the majority of the cases, are border line lesions included in recectable lesions in the NCCN Guidelines?

Results 3.4 Clinical Outcomes

Line 200-

and two (1.4%) required a different needle type due to technical failure or failure to achieve the expected diagnosis via ROSE.

One lesion was difficult to puncture and required the use of a 22G FNB needle.

The other lesion was technically successful using a 22G FNA needle although no malignant cells were observed via ROSE.

Rescue methods is well documented, but there is not enough information about technical failures in 19G TopGain. In each of the two cases, please add what kind of technical difficulty was there, whether it was the puncture itself, or whether it was the sample collection.

3.5 Adverse Events and Table5

In this paper, all cases using the 19G TopGain Needle are analyzed for adverse events. However, there is a possibility that the safety of 19G TopGain is overestimated because there are cases in which other puncture needles such as 22G needles, which are presumed to be safer, were used in combination.

In order to accurately evaluate the safety of puncture with the 19G TopGain Needle, only 146 lesions without other puncture needles should be analyzed.

Discussion Line 301-

In low-cellularity tumors, such as pancreatic cancer, only 302 interstitial components may be collected and the area of the tumor in the specimen may 303 be insufficient, even when the target lesion has been successfully punctured.

Can you show the pathological findings that only the interstitial components could be collected with FNA needle or 22G FNB needle and the tumor could be collected with FNB with 19G TopGain in actual pancreatic cancer cases?

Author Response

RESPONSE TO REVIEWER 3

We wish to thank the reviewer for the detailed review of our manuscript and valuable comments. We have responded to the comments and highlighted the changes made in the revised manuscript based on each comment.

  1. Introduction Line 79-82

    The 19G TopGain needle is expected to be as easy to use as the 22G FNB needle (Figure 1).
    As the author himself states in the discussion,RCT of the third-generation 19G and 22G FNB needles is required to evaluate the puncture performance of the 19G TopGain needle. Please state only the fact.
    The 3rd generation Franseen FNB needle, TopGain, is made of stainless steel, which is more flexible than other 3rd generation FNB needles made of nitinol or cobalt chrome, for the purpose of widening the penetration angle of the needle.

    Response: Thank you for raising this valid concern. As you pointed out, I should have said only the facts since it is an Introduction. We have corrected it as you suggested. The following corrections have been made.

    Original: TopGain, a third-generation Franseen FNB needle, is made of stainless steel and has more flexibility than other third-generation FNB needles made of nitinol or cobalt chrome. The 19G TopGain needle is expected to be as easy to use as the 22G FNB needle (Figure 1).
    Revised: TopGain, a third-generation Franseen FNB needle, is made of stainless steel, which is more flexible than other 3rd generation FNB needles made of nitinol or cobalt chrome, for the purpose of widening the penetration angle of the needle (Figure 1).
  2. Materials and Methods

    3. EUS-TA Procedure

    Line 109  Misprint

    suspected to be malignant prior to the procedure..
    Response: Thank you for pointing this out. We apologize for the error. We have corrected it (line 110).

    Original: suspected to be malignant prior to the procedure..
    Revised: suspected to be malignant prior to the procedure.

  3. Line 114-117

    The 22G FNB needle is used for resectable lesions, while the 22G or 25G FNA needles are used for lesions that are difficult or expected to be difficult to puncture with a 19G or 22G FNB needle due to the location or size of the lesion.

    In case of PDAC, which is the majority of the cases, are border line lesions included in recectable lesions in the NCCN Guidelines?

    Response: Thank you for your question. With regard to pancreatic cancer, the majority of our patients had unresectable pancreatic cancer. As you said, borderline resectable pancreatic cancer was examined as resectable cancer, so mainly 22G FNB needle was selected. We modified 2.3 EUS-TA Procedure(line 117) as followed.

    Original: The 22G FNB needle is used for resectable lesions,
    Revised: The 22G FNB needle is used for non-unresectable lesions lesions, such as resectable or borderline resectable pancreatic cancer,

  4. Results 3.4 Clinical Outcomes
    Line 200-

    and two (1.4%) required a different needle type due to technical failure or failure to achieve the expected diagnosis via ROSE. 

    One lesion was difficult to puncture and required the use of a 22G FNB needle. 

    The other lesion was technically successful using a 22G FNA needle although no malignant cells were observed via ROSE.

    Rescue methods is well documented, but there is
    not enough information about technical failures in 19G TopGain. In each of the two cases, please add what kind of technical difficulty was there, whether it was the puncture itself, or whether it was the sample collection.

    Response: Thank you for this valid comment. As you pointed out, the details of the case in which the needle was switched from 19G TopGain to another needle were not clear. The following corrections have been made in section 3.4, Clinical Outcome (line 211).

    Original: One lesion was difficult to puncture and required the use of a 22G FNB needle. The other lesion was technically successful using a 22G FNA needle although no malignant cells were observed via ROSE.
    Revised: One lesion was difficult to puncture by the 19G TopGain and required the use of a 22G FNB needle. Puncture in the other lesion was technically successful; however, no malignant cells were observed via ROSE. Although additional acquisition using a 22G FNA needle was performed, no malignant findings were finally obtained on histological diagnosis.

  5. 5 Adverse Events and Table5

    In this paper, all cases using the 19G TopGain Needle are analyzed for adverse events. However, there is a possibility that the safety of 19G TopGain is overestimated because there are cases in which other puncture needles such as 22G needles, which are presumed to be safer, were used in combination.

    In order to accurately evaluate the safety of puncture with the 19G TopGain Needle, only 146 lesions without other puncture needles should be analyzed.

    Response: Thank you for pointing out this very important point. We believe that in order to evaluate the safety of 19G TopGain as you have pointed out, we should have focused our evaluation on cases in which only 19G TopGain was used. We have made the following correction (line 149 and 323).

    Original:Adverse events were evaluated according to the classification developed by the American Society for Gastrointestinal Endoscopy workshop and were divided into intraoperative adverse events, early adverse events (up to 14 days), and late adverse events (after 14 days) [35].
    Revised: Adverse events were evaluated only in cases where 19G TopGain alone was used according to the classification developed by the American Society for Gastrointestinal Endoscopy workshop and were divided into intraoperative adverse events, early adverse events (up to 14 days), and late adverse events (after 14 days) [35].

    Original: The early adverse event rate was 4.1% (6/147), and the rate of moderate or severe adverse events was 2.0% (3/147) (Table 5).
    Revised: The early adverse event rate was 4.1% (6/146)(95%CI, 1.70-8.87%), and the rate of moderate or severe adverse events was 2.1% (3/146)(95%CI, 0.43-6.14%) (Table 5)

  6. Discussion Line 301-

    In low-cellularity tumors, such as pancreatic cancer, only 302 interstitial components may be collected and the area of the tumor in the specimen may 303 be insufficient, even when the target lesion has been successfully punctured.

    Can you show the pathological findings that only the interstitial components could be collected with FNA needle or 22G FNB needle and the tumor could be collected with FNB with 19G TopGain in actual pancreatic cancer cases?

    Response: Thank you for your comment. There were two cases in which specimens were collected with a 22G FNB needle due to inadequate specimen volume, and the patient was transferred to a 19G FNB needle, but both cases were NETs. Therefore, we were not able to provide pathology images that met your expectations. However, we thought it necessary to compare the pathology findings of the 22G needle and the 19G needle, so we have presented the pathology findings as Figure 4.

    We have added the following to the Discussion section (line 297).

    Original: As specimens collected using the 19G FNB needle are large, the expected rate of successful analysis at CGP is high [31,41]
    Revised: As specimens collected using the 19G FNB needle are large, the expected rate of successful analysis at CGP is high [31,41] (Figure 4).

    Figure 4. The comparison of histological findings between 19G FNB needle and 22G FNB needle.